U-TSS: a novel time series segmentation model based U-net applied to automatic detection of interference events in geomagnetic field data

Shan Weifeng 1 2
Wang Mengyu 1
Xia Jinzhu 1
Chen Jun 3 shanyejunjie@163.com
Li Qi 4 darcyli@163.com
Xing Lili 1
Zhang Ruilei 1
Wang Maofa 5
Zhang Suqin 4
Zhang Xiuxia 6
1 School of Emergency Management, Institute of Disaster Prevention , Sanhe , China
2 Hebei Key Laboratory of Resource and Environmental Disaster Mechanism and Risk Monitoring , Sanhe, Hebei , China
3 Earthquake Administration of Anhui Province , Hefei , China
4 China Earthquake Administration, Institute of Geophysics , Beijing , China
5 Guilin University of Electronic Technology, Guangxi Key Laboratory of Trusted Software , Guilin , China
6 Earthquake Administration of Jiangsu Province , Nanjing , China
Kong Xiangjie
Electronic publication date: 2025 Feb 11
Publication date: 2025
Volume: 11
Electronic Location ID: e2678
Received 2024 Nov 5; Accepted 2025 Jan 10
Copyright: © 2025 Shan et al.
Copyright year: 2025
Copyright holder: Shan et al.
License: This is an open access article distributed under the terms of the Creative Commons Attribution License, which permits unrestricted use, distribution, reproduction and adaptation in any medium and for any purpose provided that it is properly attributed. For attribution, the original author(s), title, publication source (PeerJ Computer Science) and either DOI or URL of the article must be cited.
License URL: https://creativecommons.org/licenses/by/4.0/

Keywords: Time series segmentation, U-net, Artificial intelligence, Geomagnetic field observation data, High-voltage direct current interference event

Funding: National Natural Science Foundation of China 42164002 Spark Program of Earthquake Sciences XH25080C Scientific Research and Development Plan Project of Langfang Science and Technology Bureau 2023011054 This work was supported by the National Natural Science Foundation of China (42164002), the Spark Program of Earthquake Sciences (XH25080C), and the Scientific Research and Development Plan Project of Langfang Science and Technology Bureau (2023011054). The funders had no role in study design, data collection and analysis, decision to publish, or preparation of the manuscript.

==============================
With the rapid advancement of Internet of Things (IoT) technology, the volume of sensor data collection has increased significantly. These data are typically presented in the form of time series, gradually becoming a crucial component of big data. Traditional time series analysis methods struggle with complex patterns and long-term dependencies, whereas deep learning technologies offer new solutions. This study introduces the U-TSS, a U-net-based sequence-to-sequence fully convolutional network, specifically designed for one-dimensional time series segmentation tasks. U-TSS maps input sequences of arbitrary length to corresponding sequences of class labels across different temporal scales. This is achieved by implicitly classifying each individual time point in the input time series and then aggregating these classifications over varying intervals to form the final prediction. This enables precise segmentation at each time step, ensuring both global sequence awareness and accurate classification of complex time series data. We applied U-TSS to geomagnetic field observation data for the detection of high-voltage direct current (HVDC) interference events. In experiments, U-TSS achieved superior performance in detecting HVDC interference events, with accuracies of 99.42%, 94.61%, and 95.54% on the training, validation, and test sets, respectively, outperforming state-of-the-art models in accuracy, precision, recall, F1-score, and AUC. Our code can be accessed openly in the GitHub repository at https://github.com/wangmengyu1/U-TSS.

Introduction

As innovations in Internet of Things (IoT) technology advance, the scope and complexity of sensor data acquisition have grown, establishing it as a crucial aspect of big data technologies (Yin et al., 2020). An increasing number of devices and sensors can collect and transmit real-time data, typically in the form of time series (Silva et al., 2021). Time series data is widely applied across various fields, including finance, ecology, economics, neuroscience, and physics (Matias et al., 2021; Polge, Robert & Le Traon, 2020). In the era of big data, the growing volume of data has revealed significant limitations of traditional time series analysis methods in handling complex patterns and long-term dependencies. The advancement of deep learning technologies provides new solutions for time series analysis. By employing convolutional neural network (CNN) and recurrent neural network (RNN), deep learning models can automatically extract latent features from time series data and capture complex temporal patterns. A growing body of research suggests that deep learning methods consistently outperform traditional approaches in time series forecasting, time series segmentation (TSS), time series classification and anomaly detection tasks.

Time series segmentation involves partitioning data into non-overlapping, automatically labeled segments. The primary objective of time series segmentation is to identify and delineate change points or event boundaries within the time series. This facilitates the organization of similar or related data segments while isolating dissimilar segments. To enable deep learning models to perform effective time series segmentation, it is essential to first label the data. Much like image segmentation tasks, where pixel-wise annotations aid in learning, properly labeled data plays a crucial role in training deep learning models for time series segmentation. The methods for labeling time series segmentation can be primarily categorized into two approaches: sliding window labeling and dense labeling (Gaugel & Reichert, 2023). Figure 1 illustrates the distinction between sliding window labeling and dense labeling. Sliding window labeling involves dividing the time series data into several fixed-length subsequences, each of which is assigned a label. Although sliding window labeling has demonstrated satisfactory results in many applications, its accuracy is inherently constrained by the size of the time window and the step size. These limitations make it difficult to capture precise event boundaries, particularly for events with variable lengths, leading to coarse segmentation and reduced interpretability in datasets with complex temporal dynamics. In contrast, the dense labeling method provides a precise approach to time series segmentation that does not rely on sliding windows. By assigning labels to each time step, dense labeling provides detailed classification information, enabling it to better adapt to diverse and irregular temporal patterns and accurately capture event boundaries.

Figure 1 An illustration of the comparative analysis between sliding window labeling and dense labeling for time series segmentation.

The upper section displays a one-dimensional time series data, represented by blue and red curves, along with the corresponding ground truth. The lower section contrasts the two labeling approaches: the sliding window labeling identifies the most frequent class (red) within each window, while the dense labeling assigns labels to every individual time step, providing a detailed representation of the segment classifications.

Similar to semantic segmentation in computer vision, effective time series segmentation depends on dense labeling and robust model architectures to ensure precise classification. In dense labeling, each individual data point, whether part of a time series or an image pixel, is assigned a specific category. A pivotal architecture in semantic segmentation is the fully convolutional network (FCN) (Long, Shelhamer & Darrell, 2015), which has significantly contributed to end-to-end pixel-level predictions. Based on FCN, many advanced convolutional neural networks have been proposed, including U-net (Ronneberger, Fischer & Brox, 2015), which has been widely used in various segmentation tasks. U-net’s symmetric contracting and expanding paths form a U-shape, and the network uses skip connections to combine positional and semantic information. Segnet (Badrinarayanan, Kendall & Cipolla, 2017) is similar to the U-net network, but it uses indexing for up-sampling to better preserve the boundary feature information. Deeplab (Chen et al., 2017) used dilated convolution and fully connected conditional random fields to improve the segmentation accuracy for image segmentation. Recent research has shown that all of these methods have been quite successful in the field of image segmentation.

Despite the advancements in segmentation techniques for images data, the inherent complexity and unique characteristics of time series data in other fields necessitate specialized segmentation methods. For instance, sleep staging and human activity recognition (HAR) are two major areas of study within time series segmentation, offering a wealth of experimental results and research insights (Yu et al., 2019). Phan et al. (2019) introduced SeqSleepNet, a hierarchical RNN designed for sleep staging, which enables end-to-end training of the network and classifies each time step of the time series to generate an output label sequence. Supratak et al. (2017) developed DeepSleepNet, a model that employs CNN to extract temporal features and utilizes bidirectional long short-term memory networks (BiLSTM) to automatically learn the transition rules between sleep stages from electroencephalogram signals. Perslev et al. (2019) proposed U-Time, a fully feed-forward deep learning algorithm for studying physiological time series segmentation of sleep data. U-Time classifies each time point in the input signal and aggregates them at fixed intervals to produce a final prediction. U-Sleep (Perslev et al., 2021) is an extension of U-Time designed for physiological time series segmentation applications. U-Sleep enables the marking of sleep stages at shorter intervals and facilitates automatic sleep staging. Zhang et al. (2024) developed a deep learning method based on 1D-CNN for human activity recognition, evaluating the model using three public datasets, all of which yielded satisfactory results. Dua, Singh & Semwal (2021) employed an end-to-end model for automatic feature extraction and activity classification. This model, which combines CNN and gated recurrent units (GRU), demonstrated robust classification performance across three publicly available human activity recognition datasets.

Similar to these physiological signal processing tasks, the detection of interference events in geomagnetic field observation data also requires precise time series segmentation methods. These geomagnetic observation data are primarily collected by geomagnetic observation instruments. Geomagnetic observation instruments can be considered as a type of IoT-like device. Compared with typical IoT devices, these instruments are larger in size, but they also have network functions, enabling them to upload the collected data to a server for further analysis. These instruments record the different components of the local magnetic field, where D represents magnetic declination, H the horizontal component, and Z the vertical component of the magnetic field. Geomagnetic observation data plays a crucial role in the study of geomagnetic storms and seismic-magnetic relationships (Beggan et al., 2024; Divett et al., 2017, 2020, 2018). However, as modern infrastructure expands, various sources of interference, including highways, subways, and high-voltage direct current (HVDC) transmission lines, have introduced significant undesired noise into geomagnetic data collection. If these interference events are not detected and preprocessed, geomagnetic field observation data cannot be applied seismic-magnetic relationship research. Among various interference events, HVDC interference events have a wide range of impacts and high frequency. Figure 2 illustrates the distribution of geomagnetic stations across the country and their impact from HVDC interferences, with stations affected by HVDC interference highlighted in red and those unaffected marked in blue. The map clearly shows that the majority of stations in China are influenced by HVDC interference events. These HVDC interference events are not related to geomagnetic activity but are primarily caused by the operation of HVDC transmission systems. During routine operation, HVDC transmission systems induce unbalanced currents in the transmission lines, generating additional magnetic fields that superimpose on the natural geomagnetic field. These unbalanced currents often arise from system faults, testing procedures, or experimental activities. The closer the observation station is to the HVDC transmission line, the more pronounced the interference. The HVDC interference events manifests as staircase-like patterns in the Z-component of the geomagnetic field, with varying durations, amplitudes, and orientations, as shown in Fig. 3. The variability and unpredictability of this interference complicate the task of accurately detecting and analyzing interference events. HVDC interference events have become a focal point in the preprocessing of geomagnetic field observation data.

Figure 2 Map of geomagnetic stations and their impact from HVDC interferences.

Red indicates stations affected by HVDV interference, blue indicates stations not affected by HVDC interference.

Figure 3 An example of HVDC interference events on instrument 1 at station 14228 on April 12th, 2017.

The red curve marked with a dotted box represents HVDC interference events. The Z-component is significantly affected by HVDC interference events, while the D and H components show no impact.

Current detection methods for HVDC interference events in geomagnetic observation data mainly rely on manual analysis and expert evaluation, typically involving the following steps. When technicians observe anomalies in the data, they first evaluate the abrupt changes in data trends, using their expertise and professional knowledge to determine whether the anomaly is a magnetic interference. Once magnetic interference are ruled out, they examine the observation system, operational records, and environmental conditions to identify other possible sources of interference. After excluding these factors, the focus shifts to analyzing changes in the Z-component. If the variation in the Z-component is significantly larger than that in the D and H components, the event is initially judged to be a HVDC interference event. Additionally, if multiple geomagnetic instruments affected by a line show the same disturbance pattern in the same time period, it is classified as HVDC interference event. However, manual detection methods often vary between individuals. To minimize such variability, identified interference events are reviewed and validated by experienced experts. The manual approach is time-consuming and labor-intensive, making it challenging to meet the demands of intensive data processing in geomagnetic observation networks and highlighting the need for an accurate and widely applicable method for detecting HVDC interference events.

In contrast to sleep staging and human activity recognition, the primary challenge in geomagnetic field observation data preprocessing is the precise detection of interference events. To address these challenges, this study draws inspiration from advancements in time series segmentation and semantic segmentation and proposes a sequence-to-sequence time series segmentation model based on U-net. We apply this model to the detection of HVDC interference events in geomagnetic field observation data, enabling the segmentation of the entire time series and the identification of HVDC interference events of varying durations at any temporal scale. The key contributions of this study are as follows: We proposed U-TSS, a novel time series segmentation model based on U-net. This model combines time series segmentation techniques with semantic segmentation to achieve high-precision dense segmentation of interference events in geomagnetic field observation data. This innovative integration effectively leverages the advantages of the U-net architecture, thereby enhancing the ability to capture complex temporal features and patterns.

The model employs a sequence-to-sequence framework, utilizing one day’s geomagnetic field observation data as input. It implicitly classifies each individual time point in the input data and aggregates these classifications into time series segments of varying lengths to produce the final predictions. This design enables the model to accurately classify each time step, significantly improving the accuracy and robustness of the dense segmentation.

By employing the Particle Swarm Optimization (PSO) algorithm (Kennedy & Eberhart, 1995) to optimize the model’s hyperparameters, the model’s performance is further enhanced. The introduction of the PSO algorithm not only accelerates the model’s convergence speed but also optimizes the selection of hyperparameters, allowing U-TSS to demonstrate higher efficiency and accuracy when processing geomagnetic field observation data.

In the remainder of the article, we outline the overall structure. “Problem Formalization” formalizes the problem. “Method” provides an overview of the U-TSS architecture and its modules. “Dataset” details the data sources and the sample production process. “Experiments” presents a series of tests conducted with state-of-the-art time series segmentation models. “Results” describes the results of the experiment. “Discussion” makes a discussion, and “Conclusions” concludes this essay.

Problem formalization

U-TSS is a sequence-to-sequence fully convolutional network designed specifically for one-dimensional time series segmentation tasks. Built upon the U-Net architecture, this model effectively addresses the limitations of traditional segmentation models when handling complex time series data. U-TSS processes time series data of arbitrary length as input, efficiently mapping the complete sequence to dense outputs in a single forward pass using fully convolutional layers. This design enhances the model’s capacity to capture global patterns in long sequences while ensuring precise segmentation at each time step, enabling accurate classification of complex temporal signals.

The problem of time series segmentation can be formally defined as follows: Let XC∈RT×C=[X1,X2,⋯,XC] represent the multivariate time series data for a specific time interval, where T denotes the number of time steps and C represents the different features of the time series data. In this study, we focus on a univariate case, thus C=1. Consequently, the univariate time series data can be expressed as a one-dimensional array X∈RT×1=[x1,x2,⋯,xT]. To address the segmentation problem, each time step t is associated with a label Yt∈RT×1, which indicates the category to which the time step belongs. This labeling approach constitutes a dense annotation method, which is essential for the segmentation task, as it enables the precise categorization of each time point according to its respective class.

In the context of geomagnetic field observation data, the input data X for the U-TSS model consists of one-dimensional time series data with the shape RT×1, corresponding to the measurements of a specific component across T time steps. The primary task is to map this input data to a sequence of predicted labels Y^t∈RT×1, producing an output for each time step. The objective is to accurately identify and localize HVDC interference events in the geomagnetic observation data, resulting in a label sequence where HVDC events are marked as 1 and the BACKGROUND is marked as 0.

This flexibility in assigning labels to every time point is a key feature of U-TSS, allowing it to handle fine-grained temporal segmentation tasks effectively. U-TSS implicitly classifies each time point in the input data and aggregates these classifications into time series segments of varying lengths. This approach enables the accurate detection of HVDC interference events in geomagnetic field observation data, thereby enhancing the accuracy and efficiency of time series segmentation methods.

Method

U-TSS model overview

To achieve precise detection of HVDC interference events in geomagnetic field observation data, U-TSS is adapted from the U-Net model, originally developed for biomedical image segmentation. The U-Net model is structured with both a contracting path and an expanding path, enabling effective feature extraction and reconstruction. The contracting path consists of four contraction stages. Each contraction stage includes two 3 × 3 convolutional layers and one 2 × 2 max pooling layer. After each contraction stage, the number of feature maps doubles while the feature maps are halved in size. The expanding path is located on the right side of the network structure and is made up of four stages. The expanding path consists of four extension stages. Each expansion stage includes one 2 × 2 up-convolutional layer and two 3 × 3 convolutional layers. The number of feature maps is reduced by half through deconvolution, and then connected to the symmetric feature maps from the contracting path on the left side. Because of the difference in size between the contracting and expanding path feature maps, the U-Net model crops the contracting path feature maps to match the size of the symmetric feature maps on the right side. The final output is obtained by applying a 1 × 1 convolutional layer to the entire model.

Figure 4 illustrates the network structure of the U-TSS model, comprising three primary modules: the contracting path, the expanding path, and a dense segmentation classifier. The contracting path focuses on feature extraction by progressively down-sampling the input time series. It captures relevant temporal patterns through convolutional operations and reduces the temporal resolution via pooling layers. The expanding path is designed to up-sampling the feature maps obtained from the contracting path, restoring the original resolution of the input. By incorporating skip connections, the model combines coarse and high-level features with fine details, which enhances its capacity for accurate time series segmentation. The dense segmentation classifier assigns a label to each time step of the input sequence. It leverages the outputs from the expanding path to produce a dense output, generating class probabilities for every time step in the time series. The design of these three modules enables the U-TSS model to effectively achieve precise segmentation and classification of time series.

Figure 4 The network structure of the U-TSS model.

The U-TSS model modifies the conventional U-Net architecture, which was originally developed for two-dimensional image data, to effectively accommodate one-dimensional time series data. Specifically, the U-TSS employs one-dimensional convolutions to effectively extract features related to HVDC interference events in geomagnetic field observation data. In contrast to two-dimensional convolution, where a sliding window operates over the feature map in both width and height directions, the one-dimensional convolution focuses solely on the width direction. This approach allows for efficient processing of the time series data, as it multiplies and sums values at corresponding positions within a single dimension. The mechanics of one-dimensional convolution are illustrated in Fig. 5. To ensure that the input feature maps of the convolution process are consistent with the output feature maps and to avoid the cropping operation during feature fusion, the U-TSS model uses SAME convolution. The convolution kernel size in Fig. 5 is three and it moves across the input sequence in fixed steps. At each step, the convolution kernel computes the output value by multiplying and summing the elements corresponding to its current position in the input region. Afterwards, the convolution kernel shifts by one step, repeating this operation until the entire input sequence is traversed. Each convolution calculation produces a point in the output feature map, resulting in a new feature map after the convolution process. Assuming that n is the current convolutional layer, the one-dimensional convolutional operation formula for this layer is shown in Eq. (1):

(1) Qjn=∑i=1K⁡Qj+i−1n−1×Wijn+bjn

where Qjn represents the output at the position j after the n-th convolutional layer, K is the size of the convolutional kernel, i is the convolution kernel index, and j represents the position index in the output feature map of the n-th layer. × denotes the convolutional operation, Wijn represents the weight of the convolution kernel at the i-th position for the j-th output in the n-th layer, and bjn is the bias corresponding to the output features at position j in layer n.

Figure 5 An example of 1-dimensional convolutional operation.

The blue region on the left side represents the input feature map, where the time point values selected by the curly brace are the elements participating in the first convolutional operation. The gray region represents the padding time points. The pink region represents the 1-dimensional convolution kernel in a time series, which slides along the input feature map, performs dot-multiplication operations with the values at corresponding positions, and sums them up to end up with the green region on the right, the output feature map.

Following the convolution operation, each output value is often transformed nonlinearly by using an activation function in order to improve the network model’s non-linear features. The activation function used in U-TSS is ReLu, which can be represented as Eq. (2):

(2) yjn=f(Qjn)=max{0,Qjn}={Qjn,Qjn≥00,Qjn < 0

where Qjn represents the input value at location j to the activation function from the convolution operation, yjn denotes the output value at location j after the activation function is applied in the n-th layer.

Contracting path

The contracting path consists of four stages, each including a max pooling layer with a pooling size of 2 and two one-dimensional convolutional layers. Max pooling preserves the most salient features by selecting the maximum value in a local region as the output value. The pooling window size is 2, moving in steps of 2 across the inputs and selecting the maximum value to pass to the next layer. Additionally, dropout layers are applied to the first and second convolutional layers in the first two contraction stages. With each contraction stage, the number of feature maps doubles while the feature maps are halved in size. The contracting path continuously performs convolution and pooling operations to obtain deep semantic information. As the data are pooled several times, the resulting low-resolution feature maps reflect the time-point semantic information, that is, each individual time-point of the geomagnetic field observation data is assigned a label.

Expanding path

In the expanding path, there are four expansion stages, each involving an up-sampling operation with a factor of 2, followed by two one-dimensional convolutional layers. In the first expansion stage, a dropout layer is introduced between the two convolutional layers. Similarly, in the third and fourth expansion stages, a dropout layer is added to each of the two one-dimensional convolutional layers. After four expansion stages, the feature maps are equal in size to the input. Throughout the continuous up-sampling process, the network can obtain deep feature information about the data. In each expansion stage, the high-resolution features of the contracting path are transferred and mixed with the up-sampled features, resulting in a doubling of the feature maps, which is the skip connection. Subsequent convolutions are then performed to capture contextual information in the encoded representations. The skip connection helps the model to successfully fuse deep semantic information with shallow positional information, achieve effective fusion of multi-scale HVDC interference events and overcome the issues of positional loss and segmentation inaccuracy in the segmentation process.

Dense segmentation classifier

The final module of the U-TSS model is the dense segmentation classifier, which assigns a label to each time step in the input sequence. Utilizing the output of the expanding path, this classifier constructs a probability distribution over the classes using softmax activation. This process results in the final predictions Y^∈RT×1, where T represents the number of time steps. By constructing this probability distribution, the dense segmentation classifier quantifies the likelihood that each observed value in the input data corresponds to an HVDC event or the BACKGROUND. This mechanism not only ensures dense segmentation by providing a label for each time step but also aggregates these classifications into segments of varying lengths, thereby allowing the U-TSS model to accurately detect HVDC interference events in the geomagnetic field observation data and pinpoint the start and end times of these events.

Dataset

Data source

The data used in this study were obtained from the Geomagnetic Network Center of China, the Institute of Geophysics, China Earthquake Administration. The data are minute-level observations derived from second-level data through Gaussian filtering. In this article, the geomagnetic field observation data and manual preprocessing logs of HVDC interference events between January 1, 2014, and December 31, 2018, were selected for the experiments. The manual preprocessing log of HVDC interference events contains the station code, instrument code, item code, and the start and end time of each HVDC event. Each station may deploy multiple geomagnetic instruments. The HVDC interference events mainly affect the Z-component of the geomagnetic field observation data, characterized by a step-like pattern. Therefore, in this study, we used the Z-component of the geomagnetic field observation data to detect HVDC interference events automatically.

HVDC sample production

The detection of HVDC interference events has primarily relied on manual inspection methods, which are often characterized by substantial time requirements and a susceptibility to human error. Furthermore, the absence of a standardized dataset for these events presents significant challenges for the advancement and validation of automated detection algorithms. Consequently, we produced a dataset specifically for the detection of HVDC interference events in geomagnetic field observation data. In this subsection, we describe the process of sample production in detail. Figure 6 illustrates the sample production process.

Figure 6 The flowchart of the HVDC sample production process.

Firstly, we selected all HVDC interference events from January 1, 2014, to December 31, 2018, from the manual preprocessing log and recorded the date, start time, end time, and affected instrument of each HVDC event. For each instrument, we individually aggregated all HVDC interference events observed on each specific day into separate records. This step ensures that each instrument’s daily records are consolidated, which is necessary for generating the label file. Missing values in the data were handled using linear interpolation.

Then, the observation data was standardized using the Z-score. In the data standardization process, the mean ( μ) and standard deviation ( σ) are computed separately for each instrument at each station on a per-day basis. The data for each instrument is standardized using the following equation:

(3) Zi=Xi−μσ

where Xi is the raw observation at time i, μ is the mean of the observation data for the corresponding instrument at that station on the given day, and σ is the standard deviation of the same instrument’s observations on that day. This standardization process does not differentiate between geomagnetically active and inactive days, and its primary aim is to normalize the data, reducing the impact of geomagnetic activity fluctuations. The normalized data is saved as a data file for the HVDC sample, named ‘station_code-instrument_code–date .npy’.

Finally, similar to the sample generation method in semantic segmentation, the U-TSS model needs to label each time point value in the geomagnetic field observation time series sample as BACKGROUND or HVDC. If a time point value is an HVDC interference event, it is labeled as 1, otherwise, it is labeled as 0. After the above operation, we get a label file with the same name as the data file of the HVDC sample, which is stored in the label folder. Therefore, each HVDC sample contains one data file and one label file, and has a consistent length.

Figure 7 shows an HVDC sample from station 12005, instrument 1, on July 3, 2017. The top half of the figure shows geomagnetic field observation data after normalization. The black line represents the BACKGROUND, and the red line indicates the HVDC interference events that occurred on that day. The bottom half of the figure shows the labels corresponding to the data, where 1 indicates HVDC, and 0 indicates the BACKGROUND.

Figure 7 An example of HVDC sample.

The dataset comprises a total of 9,255 samples, collected from 126 affected observation stations. To enhance the model’s generalization capabilities, the samples were randomly shuffled and divided into three subsets: 7,405 samples for training, 925 for validation, and 925 for testing, following an 8:1:1 ratio. The number of samples per station varies, depending on the frequency and occurrence of HVDC interference events throughout the study period.

Experiments

Experiment setting

The U-TSS model proposed in this article was developed using Python with Keras for the model design and construction. TensorFlow was chosen as the underlying deep learning library (Abadi et al., 2016). Training the model employed two NVIDIA Tesla V100 FHHL 16G GPU cards, with two Intel(R) Xeon(R) Silver 4116 CPUs @ 2.10 GHz processors and 256 GB of memory. The Adam optimizer was used (Diederik & Ba, 2014), with a batch size of 64 for the training set and a batch size of 2 for the validation set. The number of epochs was set to 300. Throughout the training process, we incorporated the early stopping mechanism, monitoring the accuracy of the model on the validation set. The PATIENCE value was set to 20, meaning that if there was no consecutive improvement in the validation set accuracy over 20 epochs, training would be immediately halted. Each epoch took approximately 16 s to train.

The model aims to minimize the loss function L(Y,Y^), which represents the distance between the predicted labels Y^ and the true labels Y. The categorical cross-entropy loss function was employed in this experiment to address the binary classification task, defined by the following formula:

(4) L(Y,Y^)=−1n∑T=1n⁡YT∗log⁡(Y^T)

where Y^T represents the predicted result of the sample and YT represents the true calculated result.

Evaluation metrics

Whether a time point value is an HVDC interference event can be viewed as a binary classification problem. In this study, evaluation metrics were accuracy, precision, recall, F1-score, and AUC.

HVDC is considered as the positive class, while BACKGROUND is considered as the negative class. True positive (TP) and true negative (TN) represent correct predictions, where the true label value and the predicted label value are the same. False positive (FP) and false negative (FN) represent incorrect predictions, where the true label value and the predicted label value are different.

Accuracy is defined as the proportion of correctly classified samples, including both HVDC and BACKGROUND data, to the total number of samples. It is calculated using the Eq. (5).

(5) accuracy=TP+TNTP+FP+TN+FN.

The precision is utilized to evaluate the model’s accuracy in predicting HVDC samples, which is defined as the proportion of correctly predicted HVDC samples to the total predicted HVDC samples. It can be calculated using Eq. (6).

(6) precision=TPTP+FP.

The recall, which is the percentage of samples predicted to be HVDC among all actual HVDC samples, is used to assess the model’s capacity to identify HVDC samples. It is calculated using Eq. (7).

(7) recall=TPTP+FN.

A harmonic mean of recall and precision is the F1-score. It gives a fair assessment by taking recall and precision into account. It is defined as shown in Eq. (8).

(8) F1−scrore=2∗precision∗recallprecision+recall.

The area under the receiver operating characteristic (ROC) curve, which is a plot of the true positive rate (TPR) against the false positive rate (FPR) for different threshold values, is referred to as the AUC (Area Under the Curve). AUC is utilized to evaluate a classifier’s performance, particularly in situations with sample imbalance. The TPR and FPR are calculated as follows, according to Eqs. (9) and (10).

(9) TPRate=TPTP+FN

(10) FPRate=FPFP+TN

Hyperparameter optimization

As the accuracy of deep learning models heavily relies on hyperparameters such as the learning rate and convolution kernel size, selecting the optimal values for these hyperparameters is a challenging task. Due to their straightforward logic structures, excellent optimization quality and efficiency, and low computational costs, numerous enhanced meta-heuristic techniques have been created in recent decades for resolving challenging optimization problems (Wang et al., 2023). As a type of meta-heuristic optimization algorithm, swarm intelligence algorithms have gained significant attention due to their ability to solve complex optimization problems. Among these, the PSO algorithm, in particular, has been widely adopted for hyperparameter tuning due to its simplicity, efficiency, and strong global optimization capabilities.

PSO operates by moving a group of ‘particles’ through a multidimensional search space to identify the optimal solution. Unlike traditional grid search and random search methods, which exhaustively evaluate predefined points or randomly chosen points in the search space, PSO iteratively refines its search based on both individual and collective intelligence, allowing for a more dynamic and efficient exploration (Band et al., 2020; Elmasry, Akbulut & Zaim, 2020; Qolomany et al., 2017). Moreover, PSO has been shown to outperform other optimization techniques, such as genetic algorithms and simulated annealing, in terms of convergence speed and computational efficiency for high-dimensional hyperparameter tuning tasks.

In this article, the primary hyperparameters of the U-TSS model, as well as those of the other comparison models (excepted TinySleepNet and 1st-difference), were optimized using the PSO algorithm. Due to the structural differences and distinct training methodology of the TinySleepNet and 1st-difference model, hyperparameter optimization could not be applied to it.

The hyperparameters for all models were set within a unified range to ensure fairness and consistency, with reference to their respective original articles. Specifically, the convolution kernel size was uniformly optimized within the range of 3 to 256, and the learning rate was constrained between 0.000001 and 0.01. The dropout rates were also optimized within the range of 0 to 0.6. For the U-TSS model, dropout rates for the first and second contraction stages in the contracting path, as well as the third and fourth expansion stages in the expanding path, were collectively referred to as dropout1, while the dropout rate for the first expansion stage was termed dropout2. This approach ensured a fair comparison while accommodating the architectural characteristics of each model. During the optimization process, the value of the fitness function gradually decreased with the increase in iterations and eventually stabilized. Ultimately, the hyperparameter optimization results for the U-TSS model are as follows: the learning rate is 0.00005, the convolution kernel size is 64, and dropout1 and dropout2 are 0.1 and 0.4, respectively. The optimization results for other comparison models are summarized in Table 1.

Table 1 Hyperparameter optimization results for U-TSS and comparison models.

The bolded row in this table indicates the optimized hyperparameters for the proposed U-TSS model.

Method	Evaluation metrics	
Hyperparameters and optimization results	
U-TSS	Learning_rate: 0.00005, kernel_size: 64, dropout 1 : 0.1, dropout 2 : 0.4	
CNN	Learning_rate: 0.00005, kernel_size: 256, dropout: 0.2	
CNN-LSTM	Learning_rate: 0.0002, kernel_size: 253, dropout: 0.6	
U-Time	Learning_rate: 0.0008, kernel_size: 11	
Encoder	Learning_rate: 0.00003, kernel_size: 16	
FCN	Learning_rate: 0.00005, kernel_size: 6	
Inception	Learning_rate: 0.00005, kernel_size: 16	
Resnet	Learning_rate: 0.001, kernel_size: 16	

Results

The accuracy and loss curves of the U-TSS model are shown in Fig. 8. During the training process, the accuracy on both the training and validation sets gradually increased, eventually reaching 99.42% and 94.61%, respectively. The U-TSS model achieved 95.54% accuracy, 88.65% precision, 76.07% recall, and an F1-score of 0.8188 on the testing set.

Figure 8 The accuracy and loss curves of the U-TSS.

(A) The accuracy curve; (B) the loss curve.

To evaluate the performance of the U-HVDC, we selected several typical detection methods for comparison. Among these, we incorporated the threshold-based first-order difference (1st-difference) method, a widely used auxiliary approach for the preliminary identification of HVDC interference events. This method relies on experts to manually define thresholds, which are subsequently verified and corrected. Additionally, we also selected some other typical time series segmentation methods to further compare the performance of U-TSS. Although there is limited literature specifically applying deep learning algorithms to detect HVDC interference events, we drew on the similarities between geomagnetic field observation data and time series data in biomedical fields, such as electrocardiogram (ECG) and electroencephalogram (EEG). Therefore, we compared U-TSS with several established segmentation algorithms commonly used in these fields, including the CNN (Acharya et al., 2017), CNN-LSTM (Oh et al., 2018), TinySleepNet (Supratak & Guo, 2020), and U-Time models. In addition, considering that geomagnetic field observation data are classical time series data, we also included several traditional time series classification methods in our comparison. These methods encompass the Encoder model (Serra, Pascual & Karatzoglou, 2018), FCN (Wang, Yan & Oates, 2017), ResNet, and Inception model (Ismail Fawaz et al., 2020). All models were trained and tested on the same datasets to assess their generalization ability.

In the comparative experiments, all models (excepted TinySleepNet and 1st-difference) were evaluated using the optimized hyperparameters obtained from the experiments. Consistent with the training approach of the U-TSS model, an early stopping mechanism was implemented with a maximum of 300 iterations and a patience of 20. Since the 1st-difference method heavily relies on threshold setting, to ensure fairness, we calculated the evaluation indicators for detecting HVDC interference events by setting different thresholds and selected the best result as the detection outcome of this method. Table 2 shows the experimental results of the 1st-difference method. A threshold of 2.0 was chosen, yielding the highest F1-score, establishing it as the optimal outcome for the 1st-difference method.

Table 2 The experimental results of 1st-difference.

The bolded row in this table represents the experiment with a first-order difference threshold of 2, which is selected for comparison with other experiments.

Evaluation metrics	
Threshold (nT)	Accuracy	Precision	Recall	F1-score	
1.0	55.33%	19.14%	73.42%	0.3037	
2.0	73.04%	24.29%	48.78%	0.3243	
3.0	78.74%	27.95%	38.22%	0.3229	
4.0	80.59%	28.73%	31.30%	0.2996	
5.0	81.93%	29.42%	25.89%	0.2754	
6.0	82.45%	28.76%	21.87%	0.2484	

Table 3 summarizes the comparative experimental results on the test set. The U-TSS model achieves an accuracy of 95.54%, a precision of 88.65%, a recall of 76.07%, an F1-score of 0.8188, and an AUC of 0.9637. In comparison, other methods show lower accuracy: the CNN model achieves 90.24%, the CNN-LSTM achieves 90.82%, and traditional methods like the 1st-difference approach achieve only 73.04%. Several other models, including TinySleepNet, U-Time, Encoder, FCN, Inception, and ResNet, report accuracies in the range of 88.96% to 90.66%, as detailed in Table 3. To provide a more intuitive comparison, the results have also been visualized in Fig. 9. In the bar chart, pink bars represent Accuracy, gray bars denote Recall, green bars indicate Precision, and blue bars correspond to the F1-Score. The results from the experiment indicate that the proposed U-TSS model achieves the highest scores across all evaluation metrics, demonstrating superior overall performance. It surpasses both traditional statistical methods and state-of-the-art deep learning approaches in detecting HVDC interference events.

Table 3 Experimental results of different models on the test set.

The bolded row in this table represents the performance metrics of the proposed U-TSS model on the test set.

Evaluation metrics	
Method	Accuracy	Precision	Recall	F1-score	AUC	
U-TSS	95.54%	88.65%	76.07%	0.8188	0.9637	
1st-difference	73.04%	24.29%	48.78%	0.3243	–	
CNN	90.24%	66.76%	52.66%	0.5888	0.8991	
CNN-LSTM	90.82%	72.86%	48.97%	0.5857	0.9006	
TinySleepNet	90.11%	83.98%	68.00%	0.7257	-	
U-Time	89.07%	79.60%	23.73%	0.3656	0.7568	
Encoder	90.66%	81.16%	38.60%	0.5232	0.8890	
FCN	88.96%	79.50%	22.63%	0.3523	0.7616	
Inception	89.30%	71.97%	31.76%	0.4407	0.7727	
Resnet	90.37%	79.05%	37.33%	0.5071	0.8139	

Figure 9 Performance comparison of different models across evaluation metrics.

Figure 10 presents the detection results of various models for an HVDC interference event in geomagnetic observation data collected on May 7, 2023, from station code 14014 and instrument code 1. The U-TSS model demonstrates superior performance, accurately detecting the interference event and precisely identifying its onset and termination. In contrast, the CNN model, while capable of detecting the event, shows less precision in delineating the event’s boundaries. The CNN_LSTM model exhibits similar detection capability to the U-TSS model but suffers from occasional false positives, as evident from the figure. Both U-Time and FCN models successfully identify the start and end times of the HVDC interference event but exhibit reduced accuracy during the event’s duration, with a noticeable number of false detections. Meanwhile, the Encoder, Inception, and ResNet models, although detecting the interference event and its boundaries, display higher false positive rates by misclassifying background data as HVDC interference. Overall, the U-TSS model outperforms all other models in terms of accuracy and reliability for detecting HVDC interference events.

Figure 10 Comparison of the performance of various models for detecting HVDC interference events in geomagnetic field data.

Discussion

Even in some complex cases, the U-TSS model still shows excellent detection performance. Figure 11 shows an example of HVDC interference events detection. The X-axis represents the corresponding time of the observation data on that day. The blue curve is the background data after the Z-score standardization, and the red curve is the HVDC events detected manually after the Z-score standardization. The orange and purple areas represent the probability that the U-TSS model predicts as the BACKGROUND and HVDC. Although the duration, direction, amplitude, and shape of each HVDC interference event are different, the U-TSS model can accurately detect all HVDC interference events, and accurately locate the start and end time of each HVDC interference event.

Figure 11 An example of the detection result of the U-TSS.

To validate the detection ability of the U-TSS model in actual HVDC interference events, we randomly selected two days of unused geomagnetic field observation data for experimentation. These days are from instrument 1 at station 14014 on May 6, 2023, and instrument 1 at station 42009 on May 28, 2023. We used the trained U-TSS model to detect HVDC interference events.

As shown in Fig. 12, there is only one HVDC interference event marked by the red curve. The U-TSS model successfully detected this HVDC event and accurately identified the start time and end time of the HVDC event.

Figure 12 The detection results of the U-TSS model for the data from instrument 1 at the station 14014 on May 6, 2023.

Figure 13 shows the detection results of the U-TSS model for the data from instrument 1 at station 42009 on May 28, 2023. There are four HVDC interference events on that day, indicated by the red curves and labeled as 1, 2, 3, and 4. The U-TSS model successfully detected events 1 and 3, while events 2 and 4 were not detected correctly. Currently, there is a certain level of false positive rate, and further improvements are needed to enhance the model’s accuracy.

Figure 13 The detection results of the U-TSS model for the data from instrument 1 at the station 42009 on May 28, 2023.

Compared to the existing manual detection technologies for HVDC interference events, the U-TSS model, as a time series segmentation method, presents several significant advantages: This model can significantly reduce the reliance on expert experience and minimize the need for manual intervention, thereby greatly decreasing labor costs. While the model achieves automatic detection of HVDC interference events with high efficiency, occasional human review may still be required to address missed or ambiguous cases, ensuring the accuracy and reliability of the detection results.

It supports the detection of HVDC interference events of different durations, varying amplitude levels, and different directions.

It can accurately locate the start time and end time of HVDC interference events.

It exhibits high detection accuracy and strong generalization capability, making it suitable for all stations without requiring separate training or optimization for each station.

Of course, the U-TSS still has some room for improvement, including: The recall of the U-TSS model still needs to be further improved, therefore, the network structure of the U-TSS model can be further optimized, such as introducing LSTM or attention mechanism.

In practical applications, subway, light rail, and HVDC interference events may occur at the same time, which may reduce the accuracy of the U-TSS model.

Conclusions

With the proliferation of IoT devices and an increasing reliance on sensor networks for real-time monitoring, the necessity for efficient processing of time series data has become paramount. As the deployment of geomagnetic field observation instruments expands and HVDC transmission lines grow, the cost and complexity associated with the manual detection of HVDC interference events are rising. To address these challenges, this article introduces U-TSS, a novel time series segmentation model based on the U-net architecture, applied to the automatic detection of HVDC interference events in geomagnetic field observation data.

U-TSS employs a fully convolutional sequence-to-sequence architecture to perform dense segmentation on one-dimensional time series data, ensuring precise labeling of each time step, and addressing the challenges posed by complex and lengthy temporal dependencies in IoT-driven applications. By implicitly classifying each time point and aggregating these classifications across varying intervals, U-TSS effectively detects events of different durations, enabling fine-grained segmentation even in the presence of complex temporal patterns. Additionally, the PSO algorithm was employed to optimize U-TSS’s hyperparameters, further enhancing its performance. Experiments demonstrate that U-TSS outperforms state-of-the-art models with accuracy, precision, recall, F1-score, and AUC values of 95.54%, 88.65%, 76.07%, 0.8188, and 0.9637, respectively, on the test set.

The key contribution of U-TSS is its ability to accurately segment and detect HVDC interference events in geomagnetic field observation data. This not only significantly reduces the labor cost and time involved in manual detection but also provides an efficient and scalable solution for handling vast amounts of time series data generated by IoT systems. In the future, we plan to extend its application to detect other types of interference events, such as those caused by subways and vehicles.

Supplemental Information

Supplemental Information 1 The network architecture of the U-TSS model.

Supplemental Information 2 Training the model and saving the best weights.

Supplemental Information 3 Calculating evaluation metrics of the model on the test set.

Supplemental Information 4 Iteratively generates batches for training.

Additional Information and Declarations

Competing Interests

The authors declare that they have no competing interests.

Author Contributions

Weifeng Shan performed the experiments, analyzed the data, performed the computation work, authored or reviewed drafts of the article, and approved the final draft.

Mengyu Wang performed the experiments, performed the computation work, prepared figures and/or tables, authored or reviewed drafts of the article, and approved the final draft.

Jinzhu Xia performed the experiments, performed the computation work, prepared figures and/or tables, authored or reviewed drafts of the article, and approved the final draft.

Jun Chen conceived and designed the experiments, authored or reviewed drafts of the article, and approved the final draft.

Qi Li conceived and designed the experiments, authored or reviewed drafts of the article, and approved the final draft.

Lili Xing conceived and designed the experiments, prepared figures and/or tables, and approved the final draft.

Ruilei Zhang analyzed the data, performed the computation work, prepared figures and/or tables, and approved the final draft.

Maofa Wang analyzed the data, prepared figures and/or tables, and approved the final draft.

Suqin Zhang analyzed the data, prepared figures and/or tables, and approved the final draft.

Xiuxia Zhang analyzed the data, prepared figures and/or tables, and approved the final draft.

Data Availability

The following information was supplied regarding data availability:

Data and code are available at GitHub and Zenodo:

https://github.com/wangmengyu1/U-TSS.

wangmengyu1. (2024). wangmengyu1/U-TSS: data and code for U-TSS (v1.0.0). Zenodo. https://doi.org/10.5281/zenodo.14507144.

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
