# Peer review of "U-TSS: a novel time series segmentation model based U-net applied to automatic detection of interference events in geomagnetic field data"

_PeerJ Computer Science, doi:10.7717/peerj-cs.2678_

## Round 0.1 · original submission · Major Revisions

This research has merits. However, the reviewers also pointed out some issues. Especially. there is a major flaw in this study which is the comparison of the behaviour on geomagnetically active days. Please carefully read the comments and revise the work accordingly.

·

Basic reporting

The language is generally clear and unambiguous. There are a few sections where it can be improved to provide clarity on the methodology. These are noted in the accompanying pdf

The scientific review is heavily weighted towards machine learning rather than the geomagnetic aspect of the research which is understandable. Those papers referenced in this section tend to be in Chinese or inaccessible, so it's not possible to judge the completeness of the review. Otherwise, the paper is well referenced.

The paper is correctly structure and follows the standard format for a scientific work.

There are several figures that can be removed without loss of clarity. These are noted in the pdf. For example Figures 3 and 9 are not overly informative. Figure 3 can be found in most U-net references and Figure 9 can be described in a sentence.

Tables 1 and 2 are very short and again could be described in the text without need for a table.

The code and data have been provided on a GitHub repository.

Some of the references are not accessible and require a DOI or definitive link as searching on the internet using the provided information failed to find them.

Experimental design

The research question is well defined and relevant. Given the impact of HVDC lines on the geomagnetic field measurements, removing their effect is an identified knowledge gap.

I suggest this is not the most rigorous investigation performed as there are implicitly assumptions regarding the optimisation of the other ML models used.

The implementation appears to be to a sufficiently high technical standard. The methods described and information provided does allow anyone to replicate the study which is an excellent output.

Validity of the findings

The experiment is designed from the point of view of a computer scientist and so looks to the positive proof only of the methodology. While the U-TSS network is excellent at finding HVDC steps in the dataset during periods of quiet geomagnetic activity, it does not appear to be checked on periods of geomagnetic storms so thus is not fully tested from the point of view of a geophyscist. I would recommend that a fuller study be done to show how the U-TSS behaves over a range of Kp values from 0 to 7 or 8 if possible.

In this sense, the experimental design is incomplete and the determination of accuracy is not correct as it only applies under very quiet conditions. How can we be sure that the false alarm rate does not rise under more active conditions. Without being too negative, the algorithm is essentially a 'step' detector and really should be compared to a standard algorithm which can pick out a dB/dt peak or some sort of rolling average filter showing when the Z value (Eq. 3) changes instantly?

I also note that the results are not well explained nor justified. i.e. the authors state the U-TSS, as trained by them, is the best and other methods are poorer but this is not a fair test really because (a) the other methods are not optimised and (b) it is not explained why the other methods would do better in any case. There is also not a comparison to simpler methods like a normalised rate of change.

Additional comments

1) I would remove the references to earthquake prediction. There is no convincing evidence that earthquakes have long period precursors and there have been no papers in the peer reviewed literature that have been generally accepted. I would suggest that you leave this out.
2) The magnetometer at Eyrewell, New Zealand suffers from similar HVDC issues. Please add a reference to their work (see pdf)
3) The magnetic data are very poorly described. Can you add a map showing where the stations are? Are they variometers or absolute measurements? Why are they so contaminated with HVDC noise? Are they minute mean cadence?
5) How do the technical people identify the HVDC effects - is it by eye or are there other tools already? Can you explain more about the manual classification process?
6) It is not clear in Equation 3 how the mean and std dev are computed. Is it once per day at each site for each instrument or some other method. What if the day is geomagnetically active?
7) Figure 15 actually shows five steps (in my examination) whereas you only identify four events, of which only two are captured by the U-TSS. Do you agree? If this happens on other days, does this alter the statistical validity of your findings?

Reviewer 2 ·

Basic reporting

No comment

Experimental design

No comment

Validity of the findings

No comment

Additional comments

This is an excellent study with valuable contributions. However, there are some areas that could be further refined to enhance clarity and completeness.
1. The abstract begins by mentioning that the collection of sensor data is a vital aspect of big data acquisition. However, the subsequent content focuses on the segmentation and processing of time series data, without explicitly clarifying the connection between sensor data and time series data.
2. The paper mentions the use of Particle Swarm Optimization (PSO) for hyperparameter optimization, but the PSO algorithm is not introduced. The paper uses the PSO algorithm to optimize the hyperparameters of U-TSS. It is recommended to add an introduction to the PSO algorithm in this paper.
3. Lines 167 to 172 should be combined into one paragraph.
4. 'T' in line 187 should be t.

Reviewer 3 ·

Basic reporting

no comment

Experimental design

no comment

Validity of the findings

no comment

Additional comments

I have read the manuscript and found that it has potential value in the automatic detection of interference events in geomagnetic field data. The proposed U-TSS model demonstrates promising results in addressing the challenges associated with time series segmentation, specifically for HVDC interference detection. However, there are still some deficiencies in the overall quality of the manuscript, both in terms of clarity and technical detail. Therefore, I propose the following suggestions for revision:
1. While dense labeling is mentioned as a key feature of the model, the distinction between dense and sliding window labeling could be further elaborated. A more in-depth explanation of why dense labeling is particularly beneficial for HVDC interference detection could strengthen the justification for choosing this approach over others.
2. The introduction mentions the use of PSO for hyperparameter optimization but does not provide a clear justification for this choice. It is essential for the authors to explain why PSO was selected over other optimization techniques such as grid search, random search, Bayesian optimization, or evolutionary algorithms.
3. In the Section 4.2 HVDC sample production, there was no mention of how missing values or outliers in the data were handled.
4. Table 3 gives the experimental results of each model on multiple indicators, but it is not intuitive. It is recommended that the author add a figure to show these experimental results.

---

## Round 0.2 · accepted · Accept

Thanks for the efforts to improve the work. The version satisfied the reviewers successfully. Congrats!

·

Basic reporting

Thanks to the authors for updating the manuscript and for providing new and clarifying experiments on the requested situations where there is higher geomagnetic activity.

Experimental design

Good

Validity of the findings

The new results are much stronger and better contextualised than the original manuscript. The methodology demonstrates a wider scope and is more applicable in general. The explanation of the need for the step detection from HVDC is also useful for motivation,

Additional comments

I would suggest you remove the references to Beggan and Divett et al - they are not needed as they do not really relate to the work in this paper.

Reviewer 2 ·

Basic reporting

no comment

Experimental design

no comment

Validity of the findings

no comment

Additional comments

The authors have adequately addressed the concerns raised. I have no further questions or requests for additional revisions.

Reviewer 3 ·

Basic reporting

no comment

Experimental design

no comment

Validity of the findings

no comment

Additional comments

I believe that this paper can be accepted.